# Dandelion Leaf Aqueous Extract Relieves Hyperuricemia and Its Complications via Modulating Uric Acid Metabolism, Renal Inflammation, and Gut Microbes

**DOI:** 10.3390/foods14223843

**Published:** 2025-11-10

**Authors:** Xiaofei Zhou, Tianxu Liu, Bingye Xu, Weiqian Zhang, Xiang Li, Fan Wei, Huan Lv, Xuemeng Ji, Bowei Zhang, Shuo Wang

**Affiliations:** 1Institute of Biomaterials and Biomedicine, School of Food and Pharmacy, Shanghai Zhongqiao Vocational and Technical University, Shanghai 201514, China; zhouxiaofei1001@163.com; 2Tianjin Key Laboratory of Food Science and Health, School of Medicine, Nankai University, Tianjin 300071, China

**Keywords:** hyperuricemia, dandelion, uric acid, xanthine oxidase, inflammation, gut microbiota

## Abstract

Dandelion is an edible and medicinal plant that has beneficial effects in various complex disorders. In this study, we investigated the regulatory effects of dandelion leaf aqueous extract (DAE) on mice with hyperuricemia (HUA) and explored its underlying mechanisms. DAE exhibited a high total phenolic content (363.31 ± 0.61 mg GAC/g) and contained 20 identified polyphenolic compounds. The administration of DAE significantly reduced serum uric acid levels and exerted protective effects on both liver and kidney function in mice with HUA. Mechanistically, DAE inhibited the NLRP3/Caspase-1 and TLR4/MyD88/NF-κB signaling pathways, leading to the downregulated mRNA expression of pro-inflammatory cytokines (IL-1β, IL-6, and TNF-α), thereby alleviating renal inflammation. Additionally, DAE modulated the gut microbiota composition and increased SCFA-producing bacteria, along with increases in fecal SCFA contents. These findings suggest that DAE effectively mitigates HUA and its associated renal complications by regulating uric acid metabolism, suppressing renal inflammation, and restoring gut microbial homeostasis. Thus, DAE holds promise as a natural adjuvant therapy for HUA and related kidney inflammation.

## 1. Introduction

Hyperuricemia (HUA) is a prevalent metabolic disorder with an increasing global incidence. It is closely associated with various diseases, including gout, chronic kidney disease, cardiovascular disorders, diabetes mellitus, nonalcoholic fatty liver disease, and metabolic syndrome [1].

Xanthine oxidase (XOD), which catalyzes the oxidation of hypoxanthine to xanthine and ultimately to uric acid, is a critical enzyme for the uric acid synthesis [2]. Elevated XOD activity leads to abnormal purine metabolism and excessive uric acid production, resulting in elevated SUA levels [3]. In addition, the insufficient elimination of uric acid is also responsible for the onset of HUA [4]. Urate disposal depends primarily on the urate transport system, including uric acid reuptake and secretory transport. The increased expression of reuptake proteins such as glucose transporter type 9 (GLUT9) enhances uric acid reabsorption, whereas the decreased expression of secretory proteins (e.g., organic anion transporter 1 (OAT1), OAT2, and ATP-binding cassette transporter G2 (ABCG2)) reduces uric acid excretion, collectively contributing to HUA [5,6].

Crystallized urate deposited in the kidney will activate the NOD-like receptor superfamily pyrin domain-containing 3 (NLRP3) inflammasome and regulate the Toll-like receptor 4 (TLR4), myeloid differentiation factor 88 (MyD88), and nuclear factor-κB (NF-κB) signaling pathways, which will lead to kidney damage [7].

Recently, several studies have posited that gut microbes are related to the onset of HUA [8,9]. Animals with HUA have altered gut microbiota composition and metabolites, short-chain fatty acids (SCFAs) in particular [10,11]. SCFAs have been reported to be beneficial to kidney function [12]. Moreover, several studies have pointed out that HUA relief is associated with the upregulation of SCFA contents [10,11]. Hence, the gut microbiota and metabolite SCFAs may be essential targets to cure HUA.

Uric acid-lowering therapies are segmented into two major categories. One is the XOD inhibitor typical of allopurinol, which inhibits the generation of uric acid via the competitive inhibition of XOD. The other is uricosuric drugs such as probenecid and benzbromarone, which promote uric acid elimination by hindering urate reabsorption in the renal tubule [13]. However, the use of these drugs is usually associated with some undesirable side effects (e.g., hepatic and renal toxicity, bone marrow suppression, and gastrointestinal irritation) [14,15]. Therefore, there is an urgent need to develop effective, low-toxicity, plant-based interventions that target multiple pathways involved in HUA pathogenesis.

Dandelion (*Taraxacum officinale* L.), a medicinal and edible plant, has been employed for treating spleen and liver complaints, kidney ailments, and gallbladder disorders for centuries [16,17]. Recently, polyphenols have been the focus of interventions for HUA and kidney inflammation [18,19,20]. Dandelion contains various polyphenols, including quercetin, luteolin, chlorogenic acid, caffeic acid, hesperidin, vanillin, etc. [21]. Meanwhile, a previous study demonstrated that dandelion root extract significantly decreased SUA [22]. However, none of the literature has focused on the uric acid-lowering and kidney inflammation relief effects of dandelion leaves. Hence, in this experiment, we sought to elucidate the role of dandelion leaf aqueous extract (DAE) against HUA and its complications and explored its underlying mechanisms, particularly regarding renal inflammation and gut microbiota regulation.

## 2. Materials and Methods

### 2.1. Materials and Reagents

Potassium oxonate (PO) was obtained from Sigma-Aldrich. Folin–Ciocalteu reagent, XOD (5 U/450 µL), xanthine, and carboxymethyl cellulose sodium salt (CMC-Na) were purchased from Solarbio Co., Ltd. (Beijing, China). Hypoxanthine (HX) was supplied by Aladdin Reagent Co. (Shanghai, China), and commercial assay kits for SUA, creatinine (Cr), blood urea nitrogen (BUN), ALT, AST, and XOD activity were procured from Jian Cheng Bioengineering Institute (Nanjing, China). Methanol and formic acid (HPLC grade) were purchased from Thermo Fisher Technology Co., Ltd. (Shanghai, China), and dried dandelion leaves were purchased from the market (Bozhou, China).

### 2.2. Preparation of DAE

Dried dandelion leaves were ground into a fine powder using a knife mill (FRITSCH, Stuttgart, Germany). Then, 50 g of the powder was extracted with distilled water at a ratio of 1:30 (*w*/*v*) for 2 h at 100 °C. The supernatant was obtained by suction filtration, concentrated by vacuum depression vaporization at 70 °C, freeze-dried, and kept at −80 °C until further use. The yield percentage of the extract was calculated as (weight of freeze-dried powder/weight of dried leaves) × 100%. The extraction yield of DAE was 29.22 ± 0.53% (*n* = 3).

### 2.3. The Measurement of the Total Phenolic Content (TPC)

The TPC of DAE was determined using the Folin–Ciocalteu colourimetric method [23], with results expressed as mg GAC equivalents per gram DAW (mg GAC/g).

### 2.4. Ultra-High-Performance Liquid Chromatography Coupled with Hybrid Quadrupole–Orbitrap Mass Spectrometry (UHPLC/Orbitrap-MS)

A total of 100 mg of the sample was extracted with 1 mL of 80% methanol (*v*/*v*) for 10 min by a vortex. After centrifugation at 12,000 rpm for 10 min, the supernatant was filtered through a 0.22 μm membrane for MS analysis. Then, 5 µL of the extract was injected into a UHPLC–Orbitrap–MS/MS system. Samples were separated using an AQ C18 column (1.8 μm, 2.1 mm × 150 mm). Mobile phase (A) was methanol, and mobile phase (B) was water containing formic acid (0.1%, *v*/*v*). A non-linear gradient was adopted (Appendix A), and the column temperature was kept at 35 °C. Mass spectra were acquired in both positive and negative ionization modes.

### 2.5. Determination of XOD Inhibition In Vitro

The inhibitory effect of DAE on xanthine oxidase activity was assessed spectrophotometrically at 295 nm, using xanthine as the substrate, according to a previously described method [24].

### 2.6. Animals and Hyperuricemia Models

Twenty-four male Kunming mice were obtained from Vital River Laboratories (Beijing, China) (SCXK (JING) 2016-0011). Animals were housed under standard laboratory conditions (24 ± 2 °C, 50 ± 5% humidity, 12 h light/dark cycle) with ad libitum access to standard rodent chow and water. All experimental procedures were strictly approved by the Ethics Committee of Tianjin University of Science and Technology.

HX and PO, dissolved in saline and CMC-Na, respectively, were applied to induce the HUA mice model. The mice were randomized and divided into a normal control group (NC, *n* = 8), a hyperuricemia group (MC, *n* = 8), and a DAE group (DAE, *n* = 8). Except for the NC group, mice were injected with HX (intraperitoneally, 300 mg/kg) and PO (intragastrically, 300 mg/kg) daily at 9:00 a.m. One hour later, mice in the DAE group received 1 g/kg of DAE by oral gavage for 19 consecutive days. Based on the body surface area conversion method, the dose is equivalent to a daily intake of 15 g of dandelion leaves for adults, which aligns with the recommendations of the Chinese Pharmacopoeia [25].

Finally, mice were deprived of food for 12 h before euthanasia, with distilled water still available. Blood samples were collected and centrifuged at 1000× *g* for 10 min at 4 °C to obtain serum. Liver and kidney tissues were excised, rinsed with cold saline, blotted dry, and weighed before being split into two parts. A part of the tissue was fixed in formalin for histopathological examination, and the rest was kept at −80 °C for further study.

### 2.7. Measurement of Biochemical Indicators in Serum and Liver of Mice

Serum was obtained to detect SUA, AST, ALT, BUN, and Cr levels. Liver tissues were homogenized to detect XOD activity. All steps were performed to abide by the manufacturer’s instructions.

### 2.8. RT-qPCR Analysis

Total RNA extraction, cDNA synthesis, and RT-PCR quantitation were performed in accordance with our previous method [26]. Primers provided by GENEWIZ (Suzhou, China) are shown in Appendix A. Relative mRNA expression was evaluated through the 2^−ΔΔCt^ method after being normalized to *β*-actin.

### 2.9. Histological Examination

Liver and kidney samples fixed with formalin were gradually dehydrated with ethanol, clarified with xylene, embedded in paraffin, sectioned, and stained with haematoxylin and eosin (H&E) for microscopy observation at 200× magnification.

### 2.10. Determination of SCFA Content

The SCFA content was detected using our previous method [26].

### 2.11. Gut Microbiota Analysis

Fecal DNA was extracted with commercial extraction kits (Magen, Guangzhou, China). The V3-V4 region of the bacterial 16S rRNA gene was amplified using a previously described method [27]. A principal coordinate analysis (PCoA) and the creation of a stacked bar plot of the bacteria composition were achieved using the R project. Circular layout representations of species abundance were generated using Circos software v.2. The KEGG pathway analysis of the OTUs was inferred using Tax4Fun v.0.69.

### 2.12. Statistical Analysis

All data were examined using a one-way analysis of variance (ANOVA) and the Duncan test (SPSS 23.0). The results are presented as mean ± standard error values of at least six repetitions unless otherwise stated.

## 3. Results

### 3.1. Determination of TPC and Identification of Polyphenol Compounds in DAE

The TPC of DAE was determined to be 363.04 ± 3.16 mg GAC/g. The phytochemical composition of DAE was further identified with a mass analysis and MS^2^ fragmentation. Figure 1 illustrates that, using the mass spectrometer, the ion flow with the positive and negative ion mode scan demonstrated better responses. Twenty polyphenol compounds in total were identified in DAE (Table 1). Among them, benzoic acid, chlorogenic acid, caffeic acid, ferulic acid, caffeoylquinic, quinic acid, tartaric acid, luteolin, and diosmetin are reported to be common in dandelion [28,29].

### 3.2. DAE Decreased SUA and Modulated Physical Signs in Mice with HUA

Compared with the NC group (34 µmol/L, Figure 2A), SUA levels significantly increased up to 282 µmol/L (*p* < 0.01) in the MC group, which further confirmed the effectiveness of the modeling method. The DAE treatment decreased the levels of SUA to 124 µmol/L (*p* < 0.01) in the mice with HUA, indicating a strong hypouricemic effect.

Body weight and organ coefficients directly reflect the health status of the mice. As shown in Figure 1B, there was no statistical difference in initial body weights in different groups. After the modeling, the weight of the mice was significantly decreased (*p* < 0.01, Figure 2B) compared with the NC group. However, the DAE treatment restored the body weight to normal levels. Additionally, the liver and kidney coefficients of the mice with HUA were increased significantly compared with the NC group (*p* < 0.01, Figure 2C,D). However, DAE administration significantly decreased the organ coefficients, reflecting improved organ health and systemic recovery.

### 3.3. DAE Alleviated Liver Injury in Mice with HUA

To further evaluate the hepatoprotective effects of DAE, ALT and AST activities were measured. A significant increase in enzyme activity was observed in the MC group (15.52, 20.67 U/L) compared with that in the NC mice (12.51, 13.84 U/L) (Figure 3A,B). As expected, DAE significantly decreased the AST and ALT activity to 9.89 and 16.30 U/L, respectively (*p* < 0.05). This result indicates that DAE alleviated the liver injury in the mice with HUA.

H&E staining was performed to further confirm the hepatic protection effect of DAE (Figure 3E). In the NC group, the hepatic lobule was centered on the central vein, surrounded by roughly radially arranged hepatocytes and hepatic blood sinusoids. The hepatocytes were round and full. The liver plate was arranged regularly and neatly, and there was no evident expansion or compression of the liver sinus. The portal area between the adjacent hepatic lobules was not abnormal. No significant inflammatory cell infiltration was observed. In the MC group, there was minimal focal necrosis of hepatocytes surrounding the central vein (black arrow) with the focal infiltration of inflammatory cells (red arrow). These pathological changes were markedly attenuated in the DAE-treated group, confirming the protective effect of DAE against HUA-induced liver damage.

### 3.4. DAE Alleviated Kidney Damage in Mice with HUA

Compared with the NC group, BUN and CR levels in the MC group were significantly increased by approximately 89.9% and 37.2%, respectively (*p* < 0.01). However, in the DAE group, we also noted a remarkable decrease in the BUN level with a reduction of 15.0% (*p* < 0.01) and the CR level with a reduction of 28.6% (*p* < 0.01) (Figure 3C,D), suggesting improved kidney function.

To further confirm the renal protection effect of DAE, we performed H&E staining of kidney sections (Figure 3F). In the NC group, the glomerulus was evenly distributed, and the number of cells and the matrix in the glomerulus were uniform. The epithelial cells of renal tubules were round, complete, and tightly arranged, and there was no apparent interstitial hyperplasia. No significant inflammatory changes were observed. In the MC group, however, there was a small amount of local perivascular connective tissue hyperplasia with a small amount of inflammatory cell infiltration (blue arrow). As expected, there was no apparent difference between the DAE and the NC groups. The results demonstrated that DAE effectively mitigates HUA-induced renal injury.

### 3.5. DAE Decreased XOD Activities Both In Vitro and In Vivo

The in vitro XOD activity was significantly decreased by DAE in a dose-dependent manner (Figure 4A). The in vivo liver XOD activity in the MC mice was 25.3% higher than the NC mice (*p* < 0.01, Figure 4B). However, DAE effectively decreased 14.9% of this (*p* < 0.01). The results indicate that DAE decreases SUA levels by inhibiting the XOD activity.

### 3.6. DAE Modulated the mRNA Expression of GLUT9, OAT1, OAT2, and ABCG2

GLUT9 is one of the most vital uric acid transporters responsible for urate reabsorption [30]. Compared with the NC group, the mRNA expression of GLUT9 exhibited a significant elevation in the MC group (*p* < 0.05, Figure 4C). However, the DAE treatment restored the mRNA level in the NC group (*p* < 0.05). OAT1, OAT2, and ABCG2 are common uric acid excretion proteins [31]. Transcription expressions of renal OAT1, OAT2, and ABCG2 were significantly reduced compared with the NC group (*p* < 0.01) (Figure 4D–F). However, the treatment with DAE partly restored their transcription levels (*p* < 0.05). ABCG2 is also expressed in the gut, especially in the ileum [6]. Hence, the mRNA expression of ileal ABCG2 was detected. It exhibited the same tendency as the results of the renal ABCG2 expression (Figure 4G). These results demonstrate that DAE lowered SUA levels by regulating the mRNA expression of uric acid transporters.

### 3.7. DAE Decreased the mRNA Expression of Renal Inflammation Cytokines and Inhibited the TLR4/MyD88/NF-κB and the NLRP3/Caspase-1 Signaling Pathways in Mice with HUA

In the present study, renal transcriptional expressions of inflammatory cytokines were determined. Figure 5A–C show that compared with the NC group, renal IL-6, IL-1β, and TNF-α mRNA expressions were significantly elevated in the MC group (*p* < 0.01). As expected, the mRNA levels of these inflammatory cytokines significantly decreased and were similar to the NC group after the DAE treatment (*p* < 0.05).

TLR4/MyD88/NF-κB and NLRP3/Caspase-1 are critical signaling pathways closely associated with inflammation, mediating inflammatory cytokines such as IL-6, TNF-α, and IL-1β. Figure 5D–H demonstrate that the mRNA expression of NLRP3, Caspase-1, TLR4, MyD88, and NF-κB was significantly increased in the MC group compared with the NC group (*p* < 0.01). Instead, DAE significantly reduced the mRNA expression of all these inflammation factors (*p* < 0.05). These findings suggest that DAE alleviates HUA-induced kidney injury by regulating the NLRP3 pathway and TLR4/MyD88/NF-κB pathway, thereby decreasing the mRNA expression of inflammatory cytokines in mice with HUA.

### 3.8. DAE Modulated Gut Microbe Disorder in Mice with HUA

16S rRNA sequencing was used to determine the impact of DAE supplementation on the gut microbiota of mice with HUA. The principal coordinate analysis (PCoA) analysis indicated that samples from different groups exhibited distinctive microbiota structures (Figure 6A). At the phylum level, Firmicutes and Bacteroidetes made up over 90% of the microbiota (Figure 6B). The relative abundance of Firmicutes and Proteobacteria significantly increased, and Bacteroidetes were significantly reduced in the MC group when compared with the NC group. By contrast, the DAE treatment significantly reversed these changes (*p* < 0.05, Appendix A). At the genus level, *Lactobacillus* accounted for the most (Figure 6C). The relative abundances of *Alloprevotella*, *Alistipes*, and *Bacteroides* were reduced, while *Lactobacillus*, *Helicobacter*, and *Lachnospiraceae UCG-006* were increased in the MC group. However, DAE treatment also reversed these changes (Appendix A). The linear discriminant analysis (LDA) and effect size (LEfSe) analysis further identified 23 influential taxonomic clades in total. In particular, *Lactobacillus* and its related species were enriched in the MC group (Figure 6D,E).

In the Tax4Fun functional analysis, the KEGG pathway related to nucleotide metabolism, specifically purine metabolism, was increased after modeling, while the treatment with DAE reversed this effect (Figure 6F,G).

### 3.9. DAE Upregulated SCFA Content in Mice with HUA

Compared with the NC group, the contents of acetic acid, propionic acid, and butyric acid in the MC group were significantly reduced by 27.2%, 31.3%, and 37.4%, respectively. After the administration of DAE, the contents of three SCFAs increased (Figure 7).

### 3.10. Correlation Analysis of Gut Microbiota with SCFAs and HUA-Related Indicators

To further understand the association between the bacterial genus and related indicators, a Pearson correlation analysis was performed (Figure 8). The results showed that *Lactobacillus*, *Muribaculum*, *Helicobacter*, and *Lachnospiraceae UCG-006* were positively associated with HUA-related indicators, and they were negatively associated with SCFAs. By contrast, *Alloprevotella*, *Bacteroides*, and *Alistipes* were negatively associated with HUA-related indicators and positively associated with SCFAs.

## 4. Discussion

Nowadays, HUA morbidity and its complications have surged worldwide [32]. However, therapeutic agents are sometimes limited by undesirable side effects. There is, therefore, an urgent need for safe, multi-targeted, and naturally derived interventions. In the present study, we found that DAE effectively lowered SUA levels, reduced renal inflammation, and modulated the gut microbiota in mice with HUA.

The homeostasis of SUA is primarily determined by the balance between uric acid biosynthesis and excretion [33]. XOD is a vital enzyme in purine nucleotide metabolism [34]. When the XOD activity increases, it produces excessive uric acid. Our study demonstrated that DAE, which is rich in diverse polyphenolic compounds, exhibited a significant inhibitory effect on XOD activity in vitro in a dose-dependent manner. Consistent with this, the DAE treatment also significantly decreased hepatic XOD activity in mice with HUA. This indicates that the inhibition of the XOD activity is a key mechanism underpinning the hypouricemic action of DAE. While previous studies have focused on dandelion root extracts [22], our findings highlight the distinct efficacy of the leaf extract, which is notably abundant in polyphenols with recognized XOD inhibitory and anti-inflammatory properties.

Beyond diminishing the uric acid production, enhancing urate excretion can also reduce SUA. Uric acid is expelled from the body through both renal and intestinal pathways. GLUT9 at the apical membranes of the proximal renal tubules is responsible for the uric acid reabsorption in the kidney [35]. OAT1 and OAT2, located in the basolateral membrane, and ABCG2 at the apical membrane, have been considered to be dominant secretory urate transporters [18]. Our results showed that the mRNA expression of GLUT9 was significantly elevated, while that of OAT1, OAT2, and ABCG2 was significantly reduced in the MC group, which is consistent with the results of a previous study [36]. The DAE treatment efficiently decreased the mRNA expression of renal GLUT9 and upregulated renal OAT1, OAT2, and ABCG2, thereby promoting uric acid excretion via the kidneys. One-third of the uric acid in the body is excreted through the gut. Recently, research has confirmed that ABCG2, also expressed on the apical membrane in the intestine, is vital for the intestinal excretion of uric acid [37]. In this study, DAE significantly upregulated the decreased mRNA expression of ABCG2, resulting in increased uric acid excretion from the intestine. These findings suggest that DAE reduces uric acid levels by regulating the mRNA expression of uric acid transporters in both the kidney and intestines.

The inflammatory response is a critical pathologic feature of HUA [38]. This study confirms that HUA induces renal inflammation, and DAE exerts a mitigating effect on HUA-induced kidney injury. Uric acid stimulates the NLRP3 inflammasome, leading to IL-1β secretion and the activation of innate immune responses [39]. Moreover, the NF-κB signaling pathway is also crucial in HUA-induced renal inflammation [40,41]. NF-κB modulates the expression of pro-inflammatory cytokine IL-1β via Toll-like receptor (TLR) signaling, in which the cytosolic TLR adapter protein MyD88 is involved. Han et al. reported that tuna meat oligopeptides alleviated HUA-induced renal inflammation by blocking the TLR4/MyD88/NF-κB signaling pathway and NLRP3 inflammasome. Consistent with this finding, our study showed that the relief effects of DAE on HUA-induced renal injury are also mediated directly by inhibiting both the TLR4/MyD88/NF-κB and NLRP3/Caspase-1 signaling pathways.

HUA burdens the kidney and causes renal injury [42]. As kidney function declines, less uric acid is excreted through the kidneys. Subsequently, the increased uric acid in the blood circulation will be excreted through the gut to compensate, directly inducing changes in the gut microenvironment [36]. This finding aligns with previous reports of gut microbiota dysbiosis in HUA models [8,9,10,11]. Here, our finding that Firmicutes was significantly increased and Bacteroidetes was significantly reduced in mice with HUA is in accordance with previous studies [28,43]. By contrast, the DAE treatment significantly reversed these changes. Shin et al. have pointed out that increased Proteobacteria is a potential signature of dysbiosis [44]. In the present study, the relative abundance of Proteobacteria was significantly increased in the mice with HUA, while DAE significantly decreased it. This result further indicates that mice with HUA demonstrated a disordered gut microbiota, while DAE demonstrated regulatory effects on the gut microbiota composition. At the genus level, the relative abundances of *Alloprevotella*, *Alistipes,* and *Bacteroides* were decreased, while the human pathogen *Helicobacter* was increased in the mice with HUA. These results are consistent with the findings of our previous study [19]. However, the DAE treatment increased their abundances. Chen et al. found that *Lachnospiraceae_UCG-006* was negatively connected with SCFA production [45], which is in accordance with our study. It is interesting that *Lactobacillus* was more abundant in the MC mice than the NC and DAE groups, which is the opposite of most studies’ findings but is consistent with the results of Huang [46]. What our study has in common with Huang’s animal models is the use of adenine. It has been reported that *Lactobacillus* may utilize adenine and stimulate proliferation [47]. In addition, nucleotide precursors can also promote the growth of *Lactobacillus* [48]. In line with this, the Tax4Fun analysis also showed that purine metabolism was increased in the MC group, but DAE decreased this. These results indicate that DAE influenced the abundance of *Lactobacillus* by interfering with the bioavailability of purine.

A bidirectional relationship exists between gut microbiota homeostasis and HUA [49]. In addition to the impact of HUA on the gut microbe composition, gut microbiota and their metabolites also play a critical role in the occurrence and development of HUA. Short-chain fatty acids (SCFAs) are gut microbiota metabolites. They have been reported to inhibit hepatic XO activity and reduce uric acid concentrations in vivo [50,51]. Moreover, SCFAs have been shown to be nephroprotective [12]. As SCFA-producing bacteria, *Bacteroides*, *Alistipes*, and *Alloprevotella* were reduced in the MC group. However, the DAE supplement increased the abundance of SCFA-producing bacteria (e.g., *Bacteroides* and *Alistipes*) and elevated fecal SCFA levels. Since SCFAs are known to inhibit xanthine oxidase activity and ameliorate renal inflammation, these changes are consistent with the observed reduction in SUA and renal protection.

## 5. Conclusions

In summary, this study provides comprehensive evidence for the therapeutic potential of DAE against HUA and its complications. DAE significantly lowered SUA in mice with HUA by inhibiting XOD activity, decreasing the mRNA expression of GLUT9, and increasing the mRNA expression of OAT1, OAT2, and ABCG2. Concurrently, it inhibited the mRNA expression of renal inflammatory cytokines via the inhibition of NLRP3/Caspase-1 and TLR4/MyD88/NF-κB signaling pathways, thereby alleviating the HUA-induced kidney inflammation. Moreover, DAE modulated the gut microbiota composition, enriched SCFA-producing bacteria, and elevated SCFAs levels, collectively contributing to its anti-hyperuricemic and nephroprotective effects.

## Figures and Tables

**Figure 1 foods-14-03843-f001:**
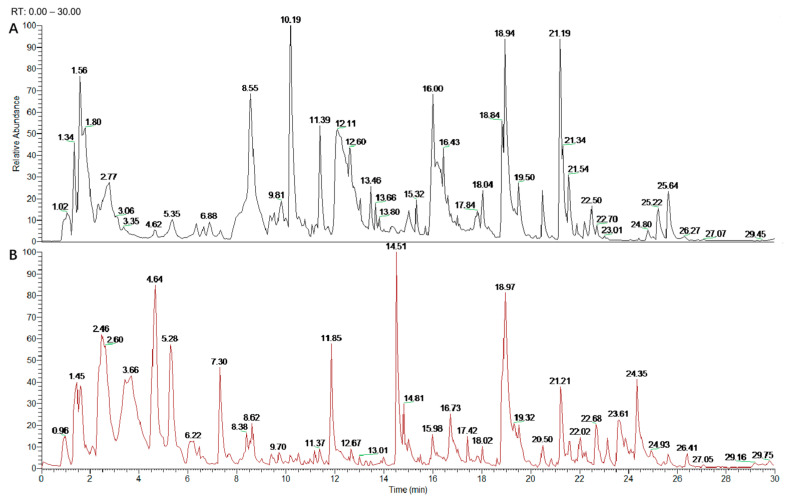
UHPLC Q-Exactive Orbitrap MS analysis of DAE phytochemical composition. Positive ion chromatogram of DAE (**A**); anion chromatogram of DAE (**B**).

**Figure 2 foods-14-03843-f002:**
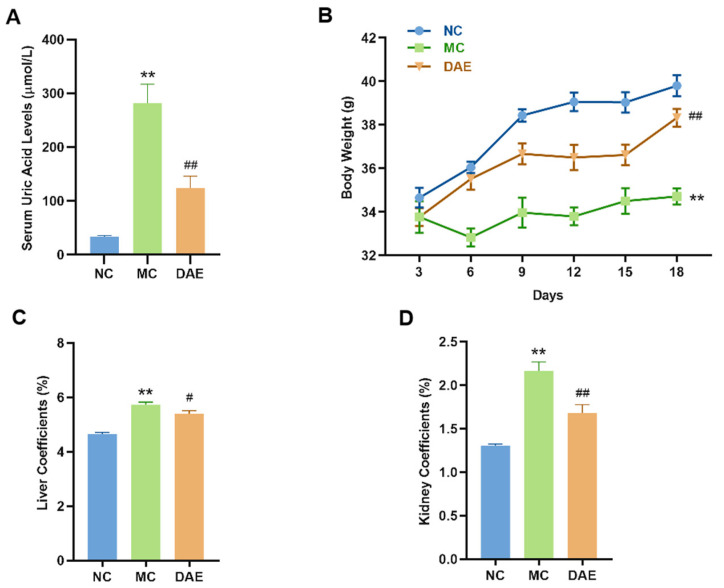
DAE reduces serum uric acid levels and modulates physiological indicators in mice with HUA: serum uric acid (**A**); body weight (**B**); liver coefficients (**C**); and kidney coefficient (**D**). * *p* < 0.05 and ** *p* < 0.01: NC group vs. MC group; # *p* < 0.05 and ## *p* < 0.01: MC group vs. DAE group.

**Figure 3 foods-14-03843-f003:**
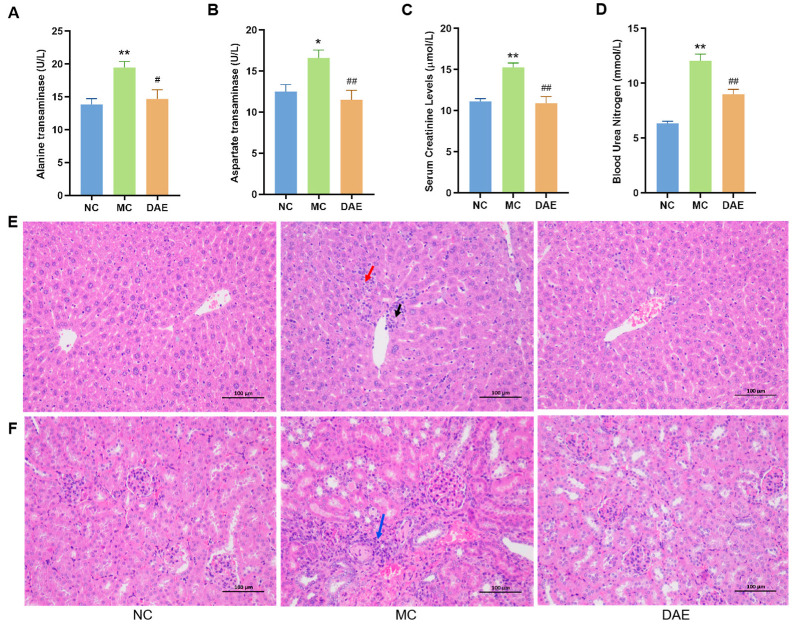
DAE alleviated hepatic and renal damage in mice with HUA: alanine transaminase (**A**); aspartate transaminase (**B**); serum creatinine (**C**); blood urea nitrogen (**D**); H&E staining of the liver (**E**); and H&E staining of the kidney (**F**) (at a magnification of 200×). * *p* < 0.05 and ** *p* < 0.01: NC group vs. MC group; # *p* < 0.05 and ## *p* < 0.01: MC group vs. DAE group. The arrows show the apparent damage (black arrow: focal necrosis of hepatocytes; red arrow: focal infiltration of inflammatory cells; and blue arrow: inflammatory cell infiltration).

**Figure 4 foods-14-03843-f004:**
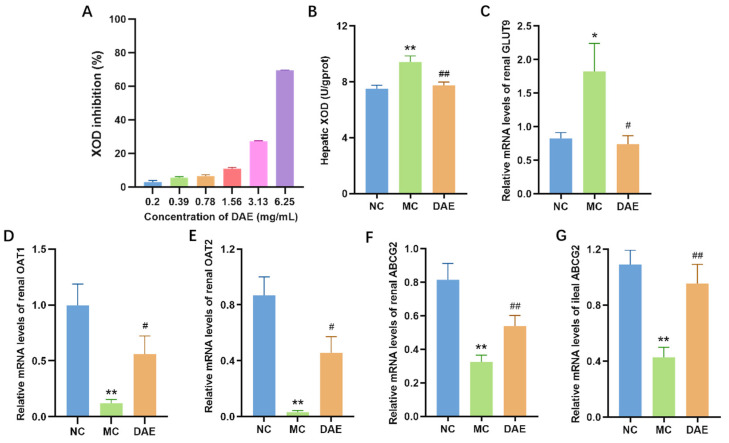
DAE decreased XOD activities and modulated mRNA expression of urate transporters in mice with HUA: XOD inhibition in vitro (**A**); hepatic XOD activities (**B**); renal GLUT9 (**C**); OAT1 (**D**); OAT2 (**E**); ABCG2 (**F**); and ileal ABCG2 (**G**). * *p* < 0.05 and ** *p* < 0.01: NC group vs. MC group; # *p* < 0.05 and ## *p* < 0.01: MC group vs. DAE group.

**Figure 5 foods-14-03843-f005:**
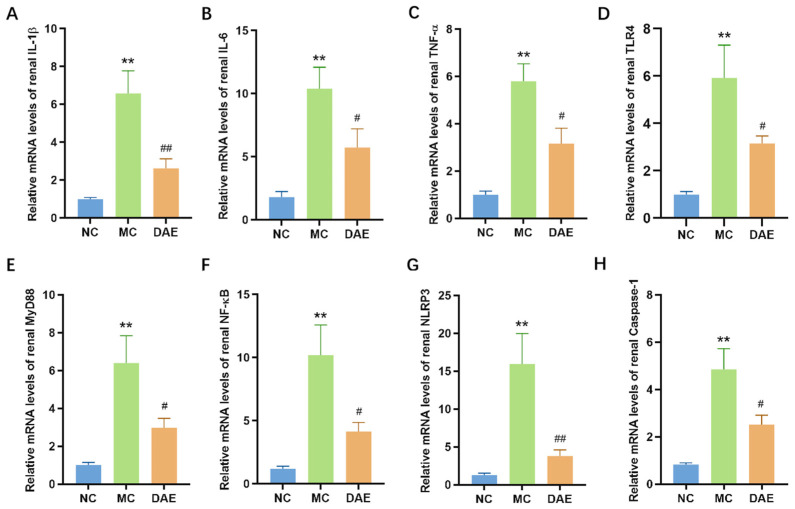
DAE reduced the levels of renal inflammation cytokines in mice with HUA. Relative mRNA expression of IL-1β (**A**); IL-6 (**B**); TNF-α (**C**); TLR4 (**D**); MyD88 (**E**); NF-κB (**F**); NLRP3 (**G**); and Caspase-1 (**H**). * *p* < 0.05 and ** *p* < 0.01: NC group vs. MC group; # *p* < 0.05 and ## *p* < 0.01: MC group vs. DAE group.

**Figure 6 foods-14-03843-f006:**
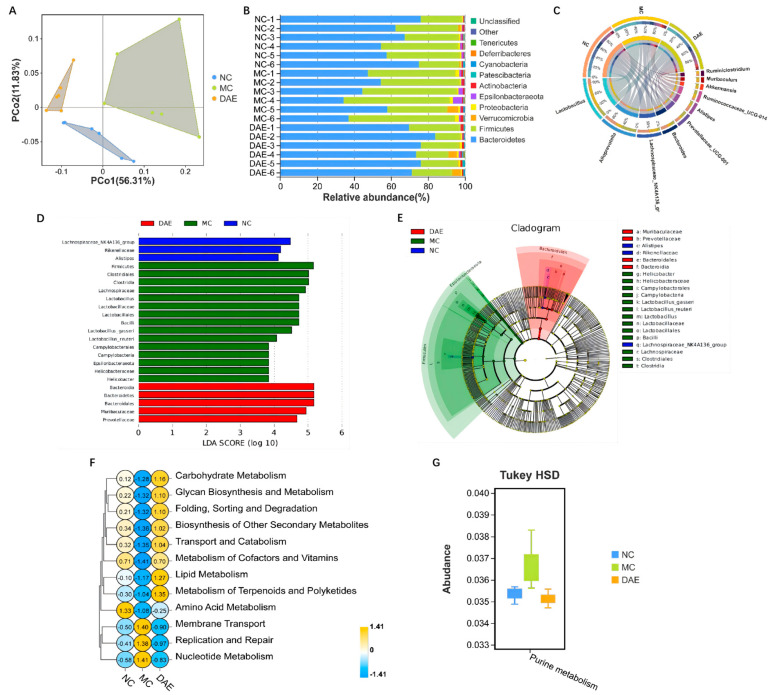
DAE modulated gut microbiota disorder in mice with HUA: PCoA diagram (**A**); gut microbiota composition at the phylum (**B**) and genus (**C**) levels; LEfSe (**D**); Cladogram (**E**); and Tax4Fun analysis according to the KEGG database at level 2 (**F**) and level 3 (**G**).

**Figure 7 foods-14-03843-f007:**
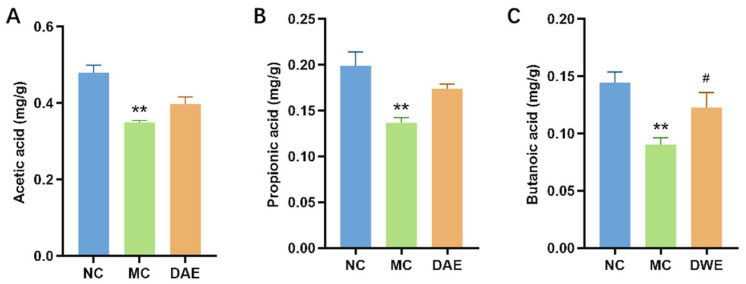
Impact of DAE on SCFAs in Mice with HUA: acetic acid (**A**); propionic acid (**B**); and butyric acid (**C**). * *p* < 0.05 and ** *p* < 0.01: NC group vs. MC group; # *p* < 0.05 and ## *p* < 0.01: MC group vs. DAE group.

**Figure 8 foods-14-03843-f008:**
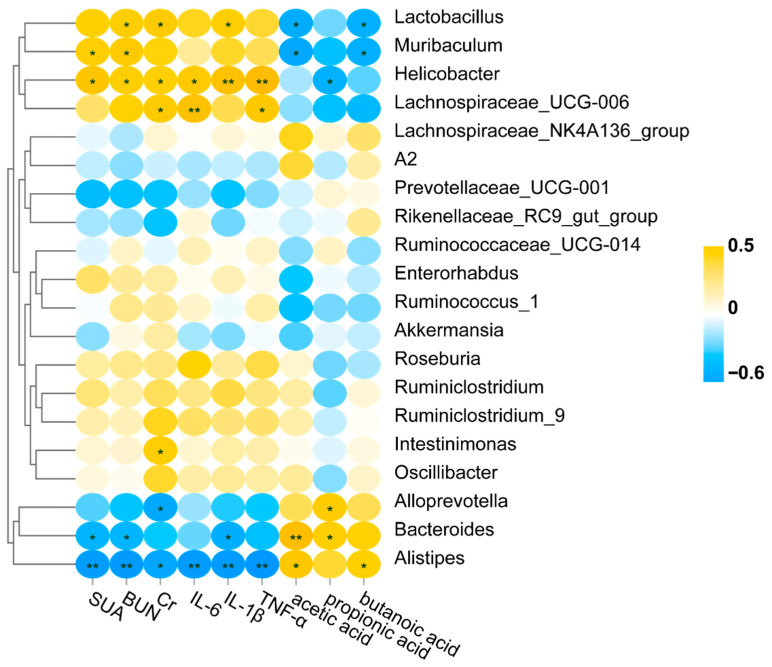
Correlation analysis of gut microbiota with SCFAs and HUA-related indicators. * *p* < 0.05; ** *p* < 0.01.

**Table 1 foods-14-03843-t001:** UHPLC/Orbitrap–MS identification of main polyphenol compounds in DAE.

Compound	Formula	Rt (min)	Ionization Mode	Calc Mw	[M − H]^−^ *m*/*z*	Delta Mass [Da]	MS/MS Fragment Ions *m*/*z*
Benzoic acid	C_7_H_6_O_2_	9.771	ESI−	122.03539	121.02811	−0.00139	93.0331; 121.0282
Caffeic acid	C_9_H_8_O_4_	10.196	ESI−	180.0412	179.03386	−0.00106	135.0439; 179.0341
Chlorogenic acid	C_16_H_18_O_9_	9.729	ESI−	354.09459	353.0874	−0.00049	85.0280; 191.0553
Cynaroside	C_21_H_20_O_11_	13.029	ESI+	448.10021	449.10748	−0.00035	153.0186; 287.0548
Quercetin-3β-D-glucoside	C_21_H_20_O_12_	12.93	ESI+	464.09545	465.10251	−0.00002	271.0250; 300.0277; 301.0356; 463.0885
Apigetrin	C_21_H_20_O_10_	13.348	ESI+	432.10518	433.11246	−0.00047	91.0544; 271.0601
Diosmetin	C_16_H_12_O_6_	15.805	ESI+	300.06308	301.07037	−0.0003	258.0514; 286.0471; 301.0697
4,5-Dicaffeoylquinic acid	C_25_H_24_O_12_	13.194	ESI−	516.12647	515.1192	−0.00031	135.0439; 173.0446; 179.0341; 353.0879
Ferulic acid	C_10_H_10_O_4_	10.299	ESI−	194.05697	193.0497	−0.00093	134.0361; 149.0600; 178.0263; 193.0500
D-(-)-Quinic acid	C_7_H_12_O_6_	9.729	ESI−	192.06242	191.05515	−0.00097	85.0280; 191.0553
Neochlorogenic acid	C_16_H_18_O_9_	8.2	ESI−	354.09468	353.08734	−0.00041	135.0440; 179.0341; 191.0553; 353.0878
Gentisic acid	C_7_H_6_O_4_	10.264	ESI−	154.02539	153.01811	−0.00122	109.0281; 153.0182
Kaempferol	C_15_H_10_O_6_	12.683	ESI+	286.04701	287.05429	−0.00072	212.1819; 287.0565
Schaftoside	C_26_H_28_O_15_	12.444	ESI+	580.1419	581.14917	−0.00092	86.8080; 287.0557; 435.8592
Luteolin	C_15_H_10_O_6_	22.725	ESI−	286.04771	285.04044	−0.00003	133.0283; 285.0406
Reynoutrin	C_20_H_18_O_11_	13.452	ESI+	434.08449	435.09171	−0.00042	69.5451; 153.0180; 303.0493
4-Hydroxyphenylacetic acid	C_8_H_8_O_3_	9.496	ESI−	152.04618	151.03886	−0.00116	71.0123; 89.0229; 101.0230; 151.0390
Biochanin A	C_16_H_12_O_5_	17.065	ESI+	284.06818	285.07538	−0.0003	55.0550; 81.0706; 242.0573; 285.0754
Apocynin	C_9_H_10_O_3_	11.199	ESI+	166.06284	167.07011	−0.00016	78.0469; 79.0547; 167.0705
3,4-Dihydroxybenzaldehyde	C_7_H_6_O_3_	8.417	ESI+	138.03163	139.03891	−0.00006	65.0394; 93.0340; 111.0444; 139.0393

## Data Availability

The original contributions presented in this study are included in the article and Appendix A. Further inquiries can be directed to the corresponding authors.

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
