# Peer review of "Dandelion Leaf Aqueous Extract Relieves Hyperuricemia and Its Complications via Modulating Uric Acid Metabolism, Renal Inflammation, and Gut Microbes"

_foods, 2025, doi:10.3390/foods14223843_

Round 1
Reviewer 1 Report
Comments and Suggestions for Authors
Methods:
- The reason for the DAE (1 g/kg) is referenced but pertains to an entirely different research domain. Require rejustification
- Absence of a positive control group, i.e., the conventional treatment for hyperuricaemia. This limits the comparative strength of the findings
- The sex of the mice not specified, which is important due to metabolic and hormonal influences on uric acid handling
Interpretation: minor speculative statements could be toned down
Writing style and grammar: several grammatical errors need to be corrected – I recommend expert proofreading. Reference formatting is inconsistent, e.g. Journal abbreviation
Figures:
- Ensure all figure panels (A, B, C, etc.) are distinctly labelled and consistent with the legend
- Scale bars and labelled should be incorporated into histological images
Comments on the Quality of English Language
As above
Author Response
Point 1: The reason for the DAE (1 g/kg) is referenced but pertains to an entirely different research domain. Require rejustification.
Response 1: Thanks for your comment. The selection of the DAE (1 g/kg) was indeed primarily based on a previous study. We acknowledge that the focus of the literature we referred to lies in liver protection, and extract type (root vs. leaf) differ from our current study. However, when we were designing this experiment, we were only able to find this single piece of literature. And this paper also describes the functional experiments conducted on mice just like us. Moreover, in this study, the administration of hot water of dandelion with 1 g/kg bw/day for 4 weeks did not possess the adverse effect. In addition, these factors are also within our consideration: Firstly, in the research on the intervention of metabolic diseases with plant extracts, a dose range of 1 g/kg is widely applicable in rodent experiments. Secondly, our in vitro results demonstrated that DAE inhibited xanthine oxidase (XOD) activity in a concentration-dependent manner (Figure 4A), supporting its potential efficacy in uric acid modulation. Given the high total phenolic content (363.31 mg GAC/g) of DAE, the 1 g/kg dose ensured a sufficient intake of polyphenol. Finally, DAE treatment significantly reduced serum uric acid (Figure 2A), improved liver and kidney function (Figure 3A–F), and restored gut microbiota balance (Figure 6A–G) without adverse effects on body weight or organ indices, confirming the appropriateness of the dose. The revisions were supplemented on Page 4, Line 160.
Point 2: Absence of a positive control group, i.e., the conventional treatment for hyperuricaemia. This limits the comparative strength of the findings.
Response 2: Thanks for your comment. There are many single-dose experiments without positive control group in animal experiments (1. Wu, Z., Huang, S., Li, T., Li, N., Han, D., Zhang, B., Xu, Z. Z., Zhang, S., Pang, J., Wang, S., Zhang, G., Zhao, J., & Wang, J. (2021). Gut microbiota from green tea polyphenol-dosed mice improves intestinal epithelial homeostasis and ameliorates experimental colitis. Microbiome, 9(1), 184. https://doi.org/10.1186/s40168-021-01115-9; 2. Hong, Y., Sheng, L., Zhong, J., Tao, X., Zhu, W., Ma, J., Yan, J., Zhao, A., Zheng, X., Wu, G., Li, B., Han, B., Ding, K., Zheng, N., Jia, W., & Li, H. (2021). Desulfovibrio vulgaris, a potent acetic acid-producing bacterium, attenuates nonalcoholic fatty liver disease in mice. Gut microbes, 13(1), 1–20. https://doi.org/10.1080/19490976.2021.1930874; 3. Quan, L. H., Zhang, C., Dong, M., Jiang, J., Xu, H., Yan, C., Liu, X., Zhou, H., Zhang, H., Chen, L., Zhong, F. L., Luo, Z. B., Lam, S. M., Shui, G., Li, D., & Jin, W. (2020). Myristoleic acid produced by enterococci reduces obesity through brown adipose tissue activation. Gut, 69(7), 1239–1247. https://doi.org/10.1136/gutjnl-2019-319114; 4. Liu, Z., Zhang, Y., Ai, C., Tian, W., Wen, C., Song, S., & Zhu, B. (2022). An acidic polysaccharide from Patinopecten yessoensis skirt prevents obesity and improves gut microbiota and metabolism of mice induced by high-fat diet. Food research international, 154, 110980. https://doi.org/10.1016/j.foodres.2022.110980; 5.).
Point 3: The sex of the mice not specified, which is important due to metabolic and hormonal influences on uric acid handling.
Response 3: Thanks for your comment. Actually, we chose male mice that were sensitive to uric acid metabolism. The revisions were supplemented on Page 4, Line 145.
Point 4: Interpretation: minor speculative statements could be toned down.
Response 4: Thanks for your comment. According to your suggestion, we have carefully reviewed the manuscript and toned down speculative statements.
Point 5: Writing style and grammar: several grammatical errors need to be corrected - I recommend expert proofreading.
Response 5: Thanks for your comment. According to your suggestion, we have thoroughly reviewed the manuscript and corrected grammatical errors, and improved sentence structure.
Point 6: Reference formatting is inconsistent, e.g. Journal abbreviation.
Response 6: Thanks for your comment. Reference formatting have been revised and checked. The revisions were supplemented on Page 17-23.
Point 7: Ensure all figure panels (A, B, C, etc.) are distinctly labelled and consistent with the legend.
Response 7: Thanks for your comment. According to your suggestion, we have made sure that all figure panels (A, B, C, etc.) are distinctly labeled and consistent with the legend. The revisions were supplemented on Page 11, Line 298.
Point 8: Scale bars and labelled should be incorporated into histological images.
Response 8: Thanks for your comment. According to your suggestion, we have added scale bars and label. The revisions were supplemented on Page 10-11, Line 273, and Line 278-279.

Reviewer 2 Report
Comments and Suggestions for Authors
This manuscript investigates the effects of dandelion leaf aqueous extract (DAE) on hyperuricemia (HUA) and related renal inflammation and gut microbiota dysbiosis in mice. The work is experimentally solid and offers an integrative view of XOD inhibition, regulation of urate transporters, suppression of inflammatory signaling pathways, and modulation of gut microbiota composition. The data are convincing and potentially valuable for developing plant-based therapeutic candidates.
However, there is issues regarding contextual accuracy, novelty framing, and comparative interpretation that need to be carefully revised.
First, the Introduction currently claims that “no literature is focused on the uric acid-lowering and kidney inflammation relief effect of dandelion.” This statement is not accurate, as previous studies have already reported that dandelion root extracts possess uricosuric and xanthine oxidase inhibitory activities in hyperuricemic rat models—for example,'Uricosuric effect of dandelion root extract on potassium oxonate–induced hyperuricemic rats'(Ukrainian Journal of Nephrology and Dialysis, 2023). These studies demonstrated that Taraxacum officinale root extracts significantly decreased serum uric acid, increased urinary uric acid excretion, and inhibited XO activity. Therefore, the manuscript should acknowledge this prior research and clarify that the novelty of the present work lies in the use of leaf aqueous extract and its integrated mechanistic exploration, particularly regarding renal inflammation and gut microbiota regulation.
Second, since there are existing studies on the root extracts, the authors are encouraged to provide either comparative data or at least a discussion explaining the biochemical and functional differences between root and leaf extracts. The discussion could mention that dandelion leaves are richer in polyphenols such as chlorogenic acid, luteolin, and caffeic acid—compounds known for their XO inhibitory and anti-inflammatory properties—while roots contain distinct phytochemicals such as inulin and sesquiterpene lactones. Including such a comparison would strengthen the scientific rationale for selecting the leaf extract and support the claim that it may provide distinct or superior benefits.
Author Response
Point 1: The Introduction currently claims that “no literature is focused on the uric acid-lowering and kidney inflammation relief effect of dandelion.” This statement is not accurate, as previous studies have already reported that dandelion root extracts possess uricosuric and xanthine oxidase inhibitory activities in hyperuricemic rat models—for example, 'Uricosuric effect of dandelion root extract on potassium oxonate–induced hyperuricemic rats'(Ukrainian Journal of Nephrology and Dialysis, 2023). These studies demonstrated that Taraxacum officinale root extracts significantly decreased serum uric acid, increased urinary uric acid excretion, and inhibited XO activity. Therefore, the manuscript should acknowledge this prior research and clarify that the novelty of the present work lies in the use of leaf aqueous extract and its integrated mechanistic exploration, particularly regarding renal inflammation and gut microbiota regulation.
Response 1: Thanks for your comment. According to your suggestion, we have acknowledged the prior research and clarify the novelty of the present work. The revisions were supplemented on Page 3, Line 91-94.
Point 2: Since there are existing studies on the root extracts, the authors are encouraged to provide either comparative data or at least a discussion explaining the biochemical and functional differences between root and leaf extracts. The discussion could mention that dandelion leaves are richer in polyphenols such as chlorogenic acid, luteolin, and caffeic acid - compounds known for their XO inhibitory and anti-inflammatory properties - while roots contain distinct phytochemicals such as inulin and sesquiterpene lactones. Including such a comparison would strengthen the scientific rationale for selecting the leaf extract and support the claim that it may provide distinct or superior benefits.
Response 2: Thanks for your comment. We have now added a comprehensive discussion comparing the biochemical composition and functional properties of dandelion leaf and root extracts in the Discussion section. The revisions were supplemented on Page 15, Line 370-378.

Round 2
Reviewer 1 Report
Comments and Suggestions for Authors
- The justification for the selection of the dose does not merit scientific merit (lines 155-156).
- Labelling on statistically significant items is still confusing in Figures 2, 4, 5 and 7. Which comparison exactly?
Author Response
Point 1: The English could be improved to more clearly express the research.
Response 1: Thanks for your comment. We have taken great care to improve the language quality of the manuscript. The text has been professionally edited for grammar, clarity, and academic style by the MDPI English Editing Service.
Point 2: The justification for the selection of the dose does not merit scientific merit.
Response 2: Thanks for your comment. Dandelion has a long history of use as a food and herbal medicine, and its consumption is generally considered safe with rare side effects. Both the FDA and many countries approve using dandelion root and extracts in dietary supplements (Faria, T.C.; Nascimento, C.C.H.C.; Vasconcelos, S.D.D.D.; Stephens, P.R.S. Literature review on the biological effects of Taraxacum officinale plant in therapy. Asian J. Pharma. Res. Devel., 2019, 7(3), 94-99.). Based on a review, the typical daily dosage range for dried dandelion roots or leaves is 4-10 g (Umi L., Jaspreet K., Kartik S., Jyoti S., Prasad R., Sawinder K., & Vishesh B. Dandelion: A Promising Source of Nutritional and Therapeutic Compounds. Recent Advances in Food, Nutrition & Agriculture. 2024, 16, 41–56.). Moreover, the recommended daily intake of dandelion is 10 to 15 grams according to the Chinese Pharmacopoeia (2020 Edition). Based on the body surface area normalization method, the dose in mice used in this study is equivalent to a daily intake of 4.4 g DAW for a 55 kg adult. Given that the extraction yield of DAE from dried leaves was 29.22%, this corresponds to roughly 15 g of dried dandelion leaves per day, which consistent with the recommended daily intake range for dandelion specified in the Chinese Pharmacopoeia. The revisions were supplemented on Page 4, Line 159-162.
Point 3: Labelling on statistically significant items is still confusing in Figures 2, 4, 5 and 7. Which comparison exactly?
Response 3: Thanks for your comment. We apologize for the confusion caused by the unclear labeling of statistical significance in Figures. The revisions were supplemented on Page 9, 10, 11, 12, and 13.

Reviewer 2 Report
Comments and Suggestions for Authors
The revisions have significantly improved the clarity and quality of the manuscript.
Author Response
Thank you for your positive feedback on our revised manuscript. We are very pleased to hear that you find the revisions have significantly improved the clarity and quality of our work. We sincerely appreciate the time and effort you have dedicated to reviewing our manuscript and providing constructive comments, which were invaluable in guiding our revisions.